# Semi-Supervised Facial Acne Segmentation Using Bidirectional Copy–Paste

**DOI:** 10.3390/diagnostics14101040

**Published:** 2024-05-17

**Authors:** Semin Kim, Huisu Yoon, Jongha Lee

**Affiliations:** AI R&D Center, lululab, Dosan Dae-Ro 318, Seoul 06054, Republic of Korea; sm.kim@lulu-lab.com (S.K.); hs.yoon@lulu-lab.com (H.Y.)

**Keywords:** acne segmentation, semi-supervised learning, bidirectional copy–paste, deep learning, semantic segmentation

## Abstract

Facial acne is a prevalent dermatological condition regularly observed in the general population. However, it is important to detect acne early as the condition can worsen if not treated. For this purpose, deep-learning-based methods have been proposed to automate detection, but acquiring acne training data is not easy. Therefore, this study proposes a novel deep learning model for facial acne segmentation utilizing a semi-supervised learning method known as bidirectional copy–paste, which synthesizes images by interchanging foreground and background parts between labeled and unlabeled images during the training phase. To overcome the lower performance observed in the labeled image training part compared to the previous methods, a new framework was devised to directly compute the training loss based on labeled images. The effectiveness of the proposed method was evaluated against previous semi-supervised learning methods using images cropped from facial images at acne sites. The proposed method achieved a Dice score of 0.5205 in experiments utilizing only 3% of labels, marking an improvement of 0.0151 to 0.0473 in Dice score over previous methods. The proposed semi-supervised learning approach for facial acne segmentation demonstrated an improvement in performance, offering a novel direction for future acne analysis.

## 1. Introduction

Facial skin disorders commonly occur among people, which is why various studies are being conducted to detect them [1,2,3,4]. Out of these, acne is a prevalent skin disorder that frequently occurs in the general population. Untreated acne has the potential to deteriorate or result in scarring. Therefore, multiple research investigations are currently being conducted to detect and classify acne based on facial images.

Figure 1 displays images of acne cropped from facial images, highlighting the variety in the color and shape of acne. Initially, acne detection primarily involved extracting features based on color or texture and utilizing classifiers. Budihi et al. [5] applied the region growing method based on pixel color similarity in facial skin images to select candidate areas for acne. Additionally, a self-organizing map was used to diagnose acne. Alamdari et al. [6] used techniques such as K-means clustering, texture analysis, and color-based segmentation to segment acne areas. Yadav et al. [7] identified candidate regions of acne presence on the face based on the hue–saturation–value (HSV) color space and then distinguished acne using classifiers trained with a support vector machine (SVM). However, these methods struggle with setting threshold values for classifying acne’s color or texture. Moreover, defining a descriptor that reflects the diverse characteristics of acne accurately is challenging for humans. Recently, methods based on deep learning have been proposed to detect acne, overcoming the aforementioned issues. Rashataprucksa et al. [8] and Hyunh et al. [9] utilized object detection models such as the faster region-based convolutional neural network (Faster-RCNN) [10] and region-based fully convolutional networks (R-FCN) [11] for acne detection. Min et al. [12] employed a dual encoder based on a convolutional neural network (CNN) and Transformer to detect face acne. Junayed et al. [13] also proposed a dual encoder based on CNN and Transformer, but they detected acne through a semantic segmentation approach. Kim et al. [14] enhanced the performance of acne segmentation by training on the positional information of acne in the final encoder.

Deep learning techniques fundamentally require labeled training data. However, labeling is time consuming and costly, and even securing original medical data, such as for acne, can be challenging. Recently, to partially address the difficulty of acquiring labeled data, much research has been conducted on semi-supervised learning, which utilizes both labeled and unlabeled data for training. Initially, various data augmentation techniques were applied to unlabeled data, employing consistency regularization [15]. Methods like Cutmix [16], which overlap parts of different images, have significantly aided in improving semi-supervised learning. Recently, a method that uses bidirectional copy–paste (BCP) alternately on the foreground and background between labeled and unlabeled data has been proposed for medical image segmentation [17]. This proposed method performs semi-supervised learning while maintaining a bidirectional relationship between the two sets of data. Additionally, to overcome the lack of training data, it was proposed to use generative models like StyleGAN2 [18,19] to create images for use in acne training [20,21].

We conducted semi-supervised learning of the acne segmentation model using BCP. We aimed to create an acne segmentation model based on semi-supervised learning using the BCP method. However, in BCP, training was conducted only with synthetic images created through copy–paste between labeled and unlabeled images. This approach made it challenging to fully reflect the characteristics of the labeled images. Although BCP performed well with computed tomography (CT) images, we discovered shortcomings in segmenting acne across various shapes and skin tones. To address this issue, we proposed adding a structure to the BCP framework that is directly trained on input labeled images. Thus, our proposed method aims to maintain the BCP structure while enabling semi-supervised learning and improving acne segmentation performance by learning from input labeled images. Additionally, in the ablation study and discussion, we conducted acne segmentation experiments using ACNE04 [22], which has similar skin tones, and analyzed issues related to this.

To verify the performance of our proposed method, we compared the acne segmentation performance with that of previous semi-supervised learning methods. For this purpose, we used images cropped primarily around acne from facial images, as illustrated in Figure 1. In this paper’s experiment, we compared the performance of acne segmentation with conventional semi-supervised learning methods by varying the proportion of labeled images during the training phase. The results showed that the proposed method achieved the highest Dice score and Jaccard index compared to previous semi-supervised methods. Notably, the superior performance over BCP demonstrates that training with both labeled images and synthetic images, rather than just synthetic images, is effective for acne segmentation applications. The main contributions of the proposed method are as follows:Fusion of labeled loss and synthetic loss: We propose a method that simultaneously calculates and fuses the labeled loss for training labeled images and the synthetic loss for training unlabeled images;Comparison of acne segmentation performance with semi-supervised learning methods: We compared the acne segmentation performance with previous semi-supervised learning methods based on our acne database and ACNE04. Additionally, through ablation studies, we compared the acne segmentation performance as the parameters of U-Net were increased.

The structure of this paper is as follows: Section 2 provides a brief introduction to acne segmentation research and semi-supervised learning. Section 3 describes the semi-supervised learning method proposed in this paper, and Section 4 presents the experimental results of the proposed method. Section 5 shows an ablation study, required when adding the training loss from input labeled images to the total loss, and Section 6 discusses the proposed method. Finally, Section 7 summarizes the conclusions of this paper.

## 2. Related Works

In this section, we briefly review previous acne detection methods based on deep learning and semi-supervised learning approaches to overcome the challenge of insufficient labeling.

### 2.1. Acne Detection

Rashataprucksa et al. [8] employed object detection models Faster-RCNN [10] and R-FCN [11] to detect acne in facial images, comparing these two models to evaluate their respective acne detection capabilities. Min et al. [12] utilized a dual encoder composed of a CNN and Transformer to extract features which were then processed through dynamic context enhancement and mask-aware multi-attention for final acne detection. Similarly, Junayed et al. [13] approached acne detection through semantic segmentation, employing a dual encoder setup with a CNN and Transformer. This method involved extracting both local and global information which was then integrated through a feature versatile block to the decoder. Kim et al. [14] segmented acne using a U-Net [23,24] structure and applied center point loss to train the last encoder on acne location information. Acne detection has primarily been conducted through either object detection or semantic segmentation. However, since the shape of acne can help differentiate the severity of the condition and assist in treatment [25], our paper aims to detect acne based on semantic segmentation.

### 2.2. Semi-Supervised Learning

Semi-supervised learning primarily applies various data augmentations to unlabeled images, focusing on consistency-based learning. FixMatch [26] extracts prediction probabilities by applying different levels of data augmentation to unlabeled data. Then, it ensures that the predictions from strongly augmented data maintain consistency based on the prediction probabilities of weakly augmented data. However, in FixMatch, only unlabeled data with predicted probabilities above a certain threshold were used for training, leading to the exclusion of many unlabeled data instances. To overcome this limitation, Full-Match [27] introduced adaptive negative learning to improve training performance. Furthermore, Wu et al. [28] applied pixel smoothness and inter-class separation in semi-supervised learning to address the blurring of pixels in edge or low-contrast areas. UniMatch [29] improved semi-supervised performance by applying stronger data augmentation in a dual structure. Notably, for strong data augmentation, Cutmix [16] was applied, which involves inserting specific parts of an image into another image. Recently, a method similar to Cutmix, BCP [17], was proposed for medical image segmentation. This method pairs labeled and unlabeled data, overlapping specific parts of their images in a manner similar to Cutmix. Semi-supervised learning is then performed using a teacher and student network structure. Loss is calculated by comparing the labeled data region with the actual ground truth, while the loss for the unlabeled data region is determined using the pseudo ground truth from the teacher network. Given BCP’s proven effectiveness in medical image segmentation, this paper aims to leverage BCP to improve facial acne segmentation performance.

## 3. Method

In this section, we provide a detailed explanation of the proposed method. First, we present the overall structure and explain the basic principles of BCP. Then, we show how the training loss is calculated in the proposed method.

### 3.1. Overall Structure

Figure 2 illustrates the overall structure proposed in this paper. It is fundamentally composed of a teacher network T and a student network S, similar to the BCP method [17]. Both T and S are constructed with U-Net [23], where the channel count of the first encoder is 16, which is the same model as BCP. Initially, T and S utilize the same pre-trained weights, $\Theta_{p}$, which are trained through supervised learning using only labeled images. However, unlike previous approaches that trained $\Theta_{p}$ with synthetic images created by applying BCP among labeled images, this study employs labeled images. This is because using labeled images to generate $\Theta_{p}$ resulted in better performance than using BCP for acne segmentation. Algorithm 1 shows the overall pseudo code.
**Algorithm 1** Training process of the proposed acne segmentation model.**Input:** labeled images
$X^{l}$, labels $Y^{l}$, unlabeled images $Y^{u}$**Output:** trained the student weights $\Theta_{s}$**Step 1:** Preparing     1.1 Setting α as a weight of unlabeled images     1.2 Setting γ as a weight of labeled loss     1.3 Setting λ as a weight of EMA update for $\Theta_{t}$     1.4 Setting η as a learning rate     1.5 Initializing $\Theta_{p}$**Step 2:** Training the pre-trained weights $\Theta_{p}$     2.1 Training and selecting the best $\Theta_{p}$          a. Computing labeled losses             $L_{i}^{l}=L_{\mathrm{seg}}\left(f\left(X_{i}^{l};\Theta_{p}\right),Y_{i}^{j}\right),\;L_{j}^{l}=L_{\mathrm{seg}}\left(f\left(X_{j}^{l};\Theta_{p}\right),Y_{j}^{l}\right)$          b. Updating $\Theta_{p}$             $\Theta_{p}=\Theta_{p}-\eta \nabla_{\Theta_{p}}\left(L_{i}^{l}+L_{j}^{l}\right)$**Step 3:** Training the student weights $\Theta_{s}$     3.1 Initializing $\Theta_{t}$ = $\Theta_{p}$, $\Theta_{s}$ = $\Theta_{p}$     3.2 Training and selecting the best $\Theta_{s}$            a. Generating synthetic images $X^{in}$ and $X^{out}$ by cropping and pasting $X^{l}$ and $X^{u}$            b. Generating pseudo GT ${\tilde{Y}}^{u}$ by $f(X^{u};\Theta_{t})$            c. Generating synthetic GT $Y^{in}$ and $Y^{out}$ by cropping and pasting $Y^{l}$ and ${\tilde{Y}}^{u}$            d. Computing synthetic losses with a mask **M**                 $L^{in}=L_{\mathrm{seg}}\left(f\left(X^{in};\Theta_{s}\right),Y^{in}\right)⊙M+\alpha \times L_{\mathrm{seg}}\left(f\left(X^{in};\Theta_{s}\right),Y^{in}\right)⊙\left(1-M\right)$                 $L^{out}=L_{\mathrm{seg}}\left(f\left(X^{out};\Theta_{s}\right),Y^{out}\right)⊙\left(1-M\right)+\alpha \times L_{\mathrm{seg}}\left(f\left(X^{out};\Theta_{s}\right),Y^{out}\right)⊙M$            e. Computing labeled losses                 $L_{i}^{l}=L_{\mathrm{seg}}\left(f\left(X_{i}^{l};\Theta_{p}\right),Y_{i}^{j}\right),\;L_{j}^{l}=L_{\mathrm{seg}}\left(f\left(X_{j}^{l};\Theta_{p}\right),Y_{j}^{l}\right)$            f. Updating $\Theta_{s}$                 $\Theta_{s}=\Theta_{s}-\eta \left(\nabla_{\Theta_{s}}\left(L^{in}+L^{out}+\gamma \left(L_{i}^{l}+L_{j}^{l}\right)\right)\right)$            g. EMA Updating $\Theta_{t}$                 $\Theta_{t}=\lambda \times \Theta_{t}+\left(1-\lambda \right)\Theta_{s}$

### 3.2. Pre-Trained Weight

In Step 3 of Algorithm 1, the weights $\Theta_{t}$ of the teacher network *T* need to be initialized with the pre-trained weights $\Theta_{p}$. The pre-trained weights are trained solely on labeled images in Step 2, which is the same as in typical supervised learning. By setting $\Theta_{p}$ as the initial values for $\Theta_{t}$ and $\Theta_{s}$, semi-supervised learning based on BCP becomes possible.

### 3.3. Bidirectional Copy–Paste for Synthetic Images

The method of generating synthetic images using BCP is as follows. First, a mask **M** of the same size as the images is created. **M** consists of zeros and ones, where the area of ones becomes the region to be copied. Then, **M** is applied to Equations (1) and (2) to generate $X^{in}$ and $X^{out}$.
(1)$$X^{in}=X_{j}^{l}⊙M+X_{p}^{u}⊙\left(1-M\right),$$
(2)$$X^{out}=X_{q}^{u}⊙M+X_{i}^{l}⊙\left(1-M\right),$$
where $X_{i}^{l}$ and $X_{j}^{l}$ are the *i*-th and *j*-th labeled images, respectively (*i* ≠ *j*), and ⊙ represents element-wise multiplication. $X_{p}^{u}$ and $X_{q}^{u}$ are the *p*-th and *q*-th unlabeled images, respectively (p≠q). **1** represents a matrix of the same size as **M**, with all elements being 1. Figure 3 provides a detailed example of a sample image generated through BCP.

### 3.4. Pseudo Synthetic Ground Truth for Supervisory Signals

The pseudo GT for unlabeled images is generated through the teacher network ***T***. First, the prediction values for the unlabeled images $X_{p}^{u}$ and $X_{q}^{u}$ are extracted using the equation below.
(3)$$P_{p}^{u}=f\left(X_{p}^{u};\Theta_{t}\right),P_{q}^{u}=f\left(X_{q}^{u};\Theta_{t}\right),$$
where *f* is a network model.

Then, as in Equation (Equation 3), a binarized pseudo GT is generated using the equation below.
(4)$${\tilde{Y}}_{p}^{u}\left(i,j\right)=\left\{\begin{array}{cc} 1 & \mathrm{if}\;P_{p}^{u}\left(i,j\right)>0.5\\ 0 & \mathrm{otherwise} \end{array}\right.,\;{\tilde{Y}}_{q}^{u}\left(i,j\right)=\left\{\begin{array}{cc} 1 & \mathrm{if}\;P_{q}^{u}\left(i,j\right)>0.5\\ 0 & \mathrm{otherwise} \end{array}\right.,$$
where *i* and *j* are coordinates.

Next, the formula below is applied to the ground truths $Y_{i}^{l}$ and $Y_{j}^{l}$ of $X_{i}^{l}$ and $X_{j}^{l}$, respectively, to generate $Y^{in}$ and $Y^{out}$, which are synthetic ground truths.
(5)$$Y^{in}=Y_{j}^{l}⊙M+{\tilde{Y}}_{p}^{u}⊙\left(1-M\right),$$
(6)$$Y^{out}={\tilde{Y}}_{q}^{u}⊙M+Y_{i}^{l}⊙\left(1-M\right).$$

### 3.5. Semi-Supervised Loss Computation

To calculate the training loss for the student network ***S***, the labeled images ($X_{i}^{l}$, $X_{j}^{l}$) and synthetic images ($X^{in}$, $X^{out}$) are each inferred through the student network ***S*** as $Q_{i}^{l}$, $Q_{j}^{l}$, $Q^{in}$, and $Q^{out}$, respectively.
(7)$$Q^{in}=f\left(X^{in};\Theta_{s}\right),\;Q^{out}=f\left(X^{out};\Theta_{s}\right),\;Q_{i}^{l}=f\left(X_{i}^{l};\Theta_{s}\right),\;Q_{j}^{l}=f\left(X_{j}^{l};\Theta_{s}\right).$$

Then, the training loss for each corresponding GT is calculated using Equation (8) through (10). $L_{\mathrm{seg}}$ is the linear combination of Dice loss and cross-entropy.
(8)$$L^{in}=L_{\mathrm{seg}}\left(Q^{in},Y^{in}\right)⊙M+\alpha \times L_{\mathrm{seg}}\left(Q^{in},Y^{in}\right)⊙\left(1-M\right),$$
(9)$$L^{out}=L_{\mathrm{seg}}\left(Q^{out},Y^{out}\right)⊙\left(1-M\right)+\alpha \times L_{\mathrm{seg}}\left(Q^{out},Y^{out}\right)⊙M,$$
(10)$$L_{i}^{l}=L_{\mathrm{seg}}\left(Q_{i}^{l},Y_{i}^{j}\right),\;L_{j}^{l}=L_{\mathrm{seg}}\left(Q_{j}^{l},Y_{j}^{l}\right),$$
where α represents the weight of the unlabeled images. The final training loss is calculated using Equation (Equation 11).
(11)$$L=L^{in}+L^{out}+\gamma \left(L_{i}^{l}+L_{j}^{l}\right),$$
where γ is a parameter that adjusts the weight of the purely supervised loss. Using the above loss, the student network ***S*** is ultimately updated. Subsequently, an Exponential Moving Average (EMA) update is performed on the teacher network ***T*** using Equation (Equation 12).
(12)$$\Theta_{t}=\lambda \times \Theta_{t}+\left(1-\lambda \right)\Theta_{s},$$
where $\Theta_{t}$ and $\Theta_{s}$ represent the parameters of the teacher network ***T*** and the student network ***S***, respectively.

## 4. Experimental Results

In this section, we analyze the performance of the proposed method for acne segmentation and compare it with previous semi-supervised learning methods. First, we describe the experimental setup and then proceed to compare the acne segmentation performance with previous semi-supervised methods.

### 4.1. Experimental Setup

To validate the performance of our proposed acne segmentation method, we acquired images of acne from facial images taken with skin diagnostic equipment [30]. Each acne image is cropped around the acne, as shown in Figure 1, and scaled to 256 × 256. We collected a total of 2000 acne images, of which 1600 were designated as the training set and the remaining 400 as the evaluation set. The optimizer used for network training was a stochastic gradient descent (SGD) with a learning rate of 0.01, momentum of 0.9, and weight decay set to 0.0001. The batch size was set to 24, comprising 12 labeled and 12 unlabeled data. Based on BCP [17], alpha was set to 0.5, and lambda was set to 0.99. The size of the area for mask 1 was set to 2/3 of the input image. Gamma was set to 0.5 according to our ablation study. Pre-training iterations were set to 10k, and semi-supervised learning iterations were set to 30k. The training evaluation was compared using Dice score and Jaccard index. All experiments were conducted on an RTX 4090, Ubuntu 20.04, Pytorch 2.1.1.

### 4.2. Comparison of Results

#### 4.2.1. Comparison between Synthetic Images and Labeled Images for Pre-Trained Weight

Pre-trained weights are trained through supervised learning. In the original approach, BCP was applied to labeled images to create synthetic images for training pre-trained weights. Our method, however, utilizes labeled images directly without applying BCP for training pre-trained weights. Thus, a comparison between these two approaches was initially conducted.

Table 1 presents the results of training with different proportions of labeled data at 3% and 7%. As shown in Table 1, generating pre-trained weights with labeled images demonstrated superiority in three metrics over using synthetic images. Therefore, we opted to train with labeled images, which, overall, provided better performance for generating pre-trained weights $\Theta_{p}$.

#### 4.2.2. Semi-Supervised Learning Comparison

We compared the semi-supervised learning performance of our proposed method with previous methods. Table 2 lists the performance of our proposed method against comparison methods across various metrics. Our method was trained in a semi-supervised manner, as proposed in Section 3, based on the pre-trained weights learned in Section 4.2.1. As shown in Table 2, our proposed method exhibited the highest performance. This suggests that training with both BCP-based synthetic images and labeled images simultaneously provided mutual benefits, leading to the superior performance of our proposed method. Figure 4 presents examples of results from each method when using 7% labeled images.

Additionally, to further compare our method with BCP, we examined the training and validation loss and the Dice score. We set the proportion of labeled images at 7% and calculated the loss using each method for comparison, as shown in Figure 5. In our method, because labeled loss is added, the initial loss is higher than BCP. However, as training progresses, it becomes lower than BCP. Especially during training, the validation loss is mostly lower than BCP. Therefore, by comparing the Dice score, we can confirm that the proposed method’s acne segmentation performance is clearly higher than that of BCP.

## 5. Ablation Study

In this section, we analyze the acne segmentation performance based on the gamma value used to fuse synthetic loss and labeled loss, and the number of parameters in U-Net, within the proposed method. Additionally, we compared the acne segmentation performance of the proposed method with BCP using the public database ACNE04.

### 5.1. Performance Variation according to γ and the Number of Channels in U-Net

In this section, we present the results of acne segmentation according to different γ values used in Equation (Equation 11) for combining synthetic image loss $\left(L^{in}+L^{out}\right)$ and labeled image loss $\left(L_{i}^{l}+L_{j}^{l}\right)$. We tested three scenarios with γ values of 0.1, 0.5, and 1.0. Generally, using γ=0.5 resulted in superior overall performance, shown in Table 3. While there were differences in some scores, high performance was demonstrated in the Dice score and Jaccard indexes, which consider overall performance. Therefore, in our experiments, we used γ=0.5.

Next, we experimented with increasing the number of channels in the first encoder of the UNet-BN used in our experiments. While BCP set the number of channels at 16, we conducted comparative experiments with increased numbers at 32 and 64. Table 4 shows the acne segmentation performance when the number of channels was increased. As the number of channels increased, performance metrics also improved. Therefore, based on using 7% labeled images for acne segmentation, compared to using BCP, the Dice score increased by 0.0424 and the Jaccard index by 0.0359.

### 5.2. Acne Segmentation Performance on ACNE04

In this subsection, we aim to compare acne segmentation performance using the public acne database ACNE04 [22]. However, ACNE04 does not have annotations for semantic segmentation. Instead, it provides bounding boxes for object detection. Using this bounding box information, we generated pseudo ground truth for semantic segmentation, as shown in Figure 6, by drawing circles passing through the center of each side of the bounding boxes. Because of the diversity of acne shapes, they do not accurately reflect the actual boundaries of acne lesions. As a result, this experiment focused on understanding the trend in performance differences with BCP rather than precise accuracy.

Originally, the ACNE04 database contained a total of 1457 images. However, some images have poor quality or contain watermarks. We excluded these and selected 1108 images that had good quality and similar shooting conditions. We selected 222 images for validation and the remaining ones for training. We cropped the selected images as described in Section 4, yielding 1600 training patches and 400 validation patches. Table 5 displays the results of semi-supervised learning using the proposed method and BCP. While the proposed method outperforms BCP, it yields a somewhat reduced improvement margin when compared to our acne database. This limitation is attributed to the crude pseudo ground truth and the characteristics of the images in ACNE04, which will be further analyzed in the Discussion section. Nevertheless, using both synthetic loss and labeled loss simultaneously helped to improve acne segmentation performance.

## 6. Discussion

The proposed method showed improved performance for our data compared to the original BCP method. However, as observed in the ablation study in Section 5, there was a decrease in acne segmentation performance improvement in ACNE04. To analyze this, we compared synthetic images composed from each dataset, as depicted in Figure 7. Our data include a variety of skin colors and diverse lighting conditions, while the ACNE04 dataset used in the experiments is predominantly composed of East Asian skin tones. Therefore, even when creating synthetic images using ACNE04, as shown in Figure 7, the images composited in the foreground exhibit less disparity and lower color gradation compared to those composed with our data. Nevertheless, actual human skin tones vary widely in color depending on race and environmental conditions. Therefore, it is expected that our method will demonstrate superior performance in actual acne segmentation compared to the original BCP approach.

## 7. Conclusions

In this paper, a semi-supervised learning method for training acne segmentation models is proposed. The original BCP method, which calculates the loss solely on synthetic images, was insufficient for detecting acne across diverse skin tones. To address this, the training process was enhanced by including original images to improve overall acne segmentation performance. The proposed method was compared with previous semi-supervised learning methods on acne patch images and demonstrated superior performance based on evaluation metrics such as the Dice score and Jaccard index. However, the performance improvement was less significant for ACNE04, where the skin color and lighting are similar. Since actual human skin color and lighting vary, performance improvements like those in the main results of this paper are expected in real applications. Future research aims to apply the proposed method across the medical imaging field to enhance performance in various areas.

## Figures and Tables

**Figure 1 diagnostics-14-01040-f001:**
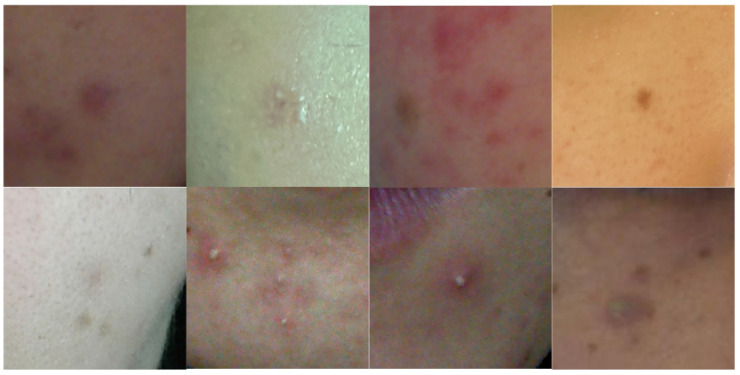
Various acne samples. The shape and skin color of acne are considerably diverse.

**Figure 2 diagnostics-14-01040-f002:**
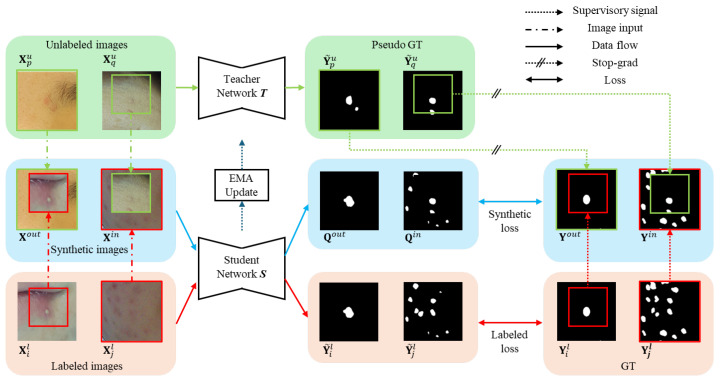
Overall structure of the proposed method. Synthetic images are generated using the bidirectional copy–paste method. The labeled image and synthetic image are each inferred for prediction values through the student network. Then, synthetic GT is generated using ground truth (GT) and pseudo GT. The training loss is calculated by sending a supervisory signal to the student network through each GT. Once the student network is trained, an EMA update is applied to the teacher network.

**Figure 3 diagnostics-14-01040-f003:**
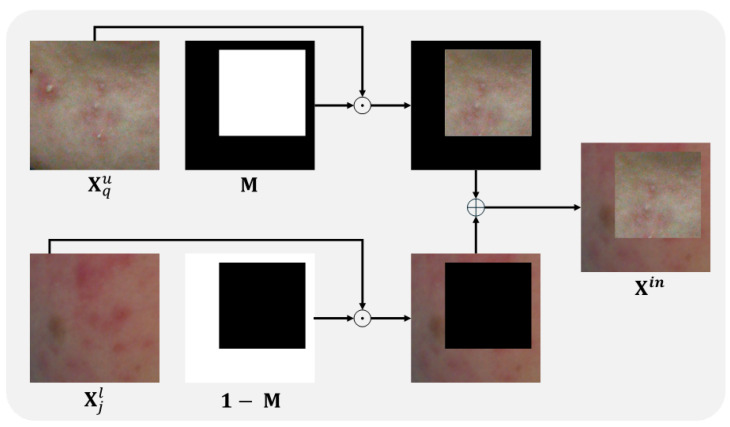
An example of creating $X^{in}$ by applying BCP to an unlabeled image $X_{q}^{u}$ and a labeled image $X_{j}^{l}$. In mask **M**, 1 represents the white area, and 0 represents the black area. **1** represents a matrix of the same size as **M**, with all elements being 1. ⊙ is element-wise multiplication, and ⊕ is element-wise addition.

**Figure 4 diagnostics-14-01040-f004:**
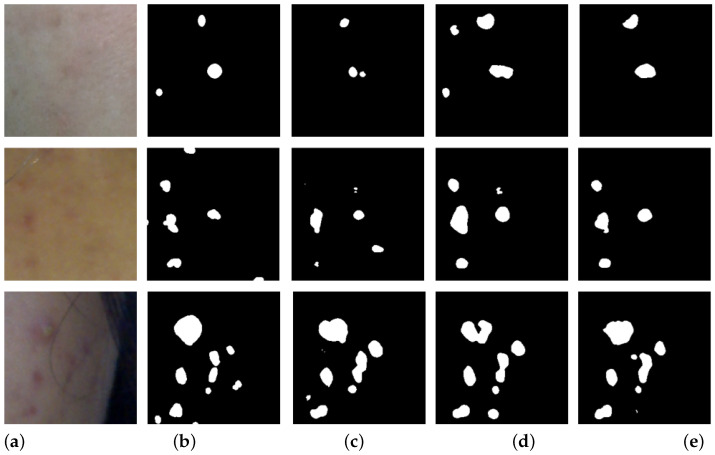
Examples of acne segmentation results from the compared semi-supervised methods. (**a**) represents the input images, (**b**) is the ground truth. (**c**) is the result of SS-Net, (**d**) is the result of BCP, and (**e**) is the result of the proposed method.

**Figure 5 diagnostics-14-01040-f005:**
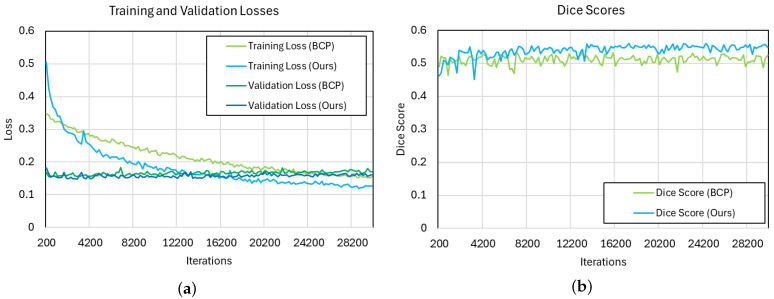
Comparison of training and validation loss and Dice score for our method and the BCP method. (**a**) shows the training and validation losses for each method, and (**b**) shows the Dice scores.

**Figure 6 diagnostics-14-01040-f006:**
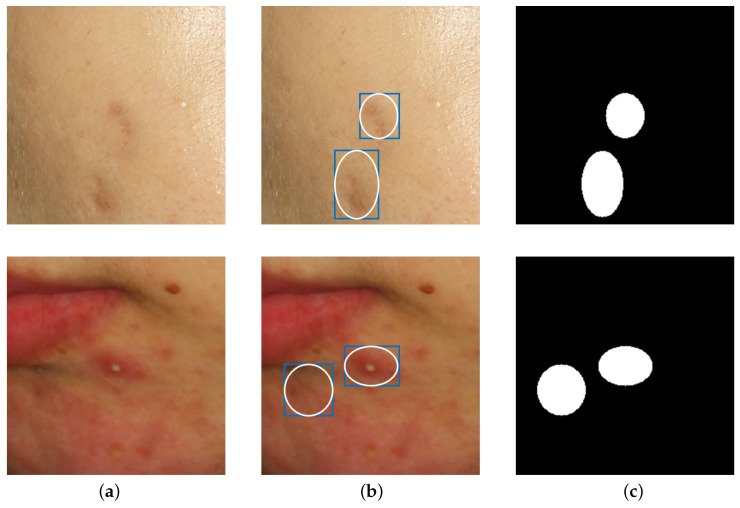
Example of creating semantic segmentation ground truth using ACNE04’s bounding boxes. (**a**) is the original image, and (**b**) shows the blue boxes indicating acne with bounding boxes. By drawing ellipses inside the bounding boxes, pseudo ground truth for acne is generated as shown in (**c**).

**Figure 7 diagnostics-14-01040-f007:**
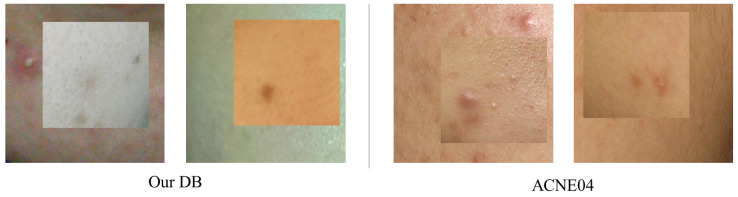
Comparison of synthetic images. The two images on the left are synthetic images generated from our database, and those on the right are from ACNE04. Our database reflects a variety of skin tones and lighting, resulting in a significant color difference between the duplicated inner images and the background images. In contrast, the ACNE04 images are synthesized in relatively similar colors.

**Table 1 diagnostics-14-01040-t001:** Comparison between synthetic images and labeled images for generating pre-trained weights $\Theta_{p}$.

Loss Type	Ratio		Metrics	
	Labeled	Unlabeled	Dice Score	Jaccard Index
Synthetic loss	3%	0%	0.4423	0.3108
Labeled loss	3%	0%	**0.4570**	**0.3203**
Synthetic loss	7%	0%	0.4784	0.3425
Labeled loss	7%	0%	**0.4951**	**0.3517**

**Table 2 diagnostics-14-01040-t002:** Comparison of acne segmentation performance of the proposed method and previous semi-supervised learning methods.

Method	Ratio		Metrics	
	Labeled	Unlabeled	Dice Score	Jaccard Index
SS-Net [28]	3%	97%	0.4732	0.3333
BCP [17]	3%	97%	0.5054	0.3617
Ours	3%	97%	**0.5251**	**0.3777**
SS-Net [28]	7%	93%	0.5162	0.3750
BCP [17]	7%	93%	0.5357	0.3912
Ours	7%	93%	**0.5603**	**0.4117**

**Table 3 diagnostics-14-01040-t003:** Comparison of acne segmentation performance based on γ.

γ	Ratio		Metrics	
	Labeled	Unlabeled	Dice Score	Jaccard Index
0.1	3%	97%	0.5177	0.3693
0.5	3%	97%	**0.5251**	**0.3777**
1.0	3%	97%	0.5205	0.3753
0.1	7%	93%	0.5522	0.4060
0.5	7%	93%	**0.5603**	**0.4122**
1.0	7%	93%	0.5588	0.4117

**Table 4 diagnostics-14-01040-t004:** Comparison of acne segmentation performance based on changes in the number of channels in the first encoder.

# Channels	Ratio		Metrics	
	Labeled	Unlabeled	Dice Score	Jaccard Index
16 (BCP [17])	3%	97%	0.5054	0.3617
16 (ours)	3%	97%	0.5251	0.3777
32 (ours)	3%	97%	0.5394	0.3912
64 (ours)	3%	97%	**0.5458**	**0.3965**
16 (BCP [17])	7%	93%	0.5357	0.3912
16 (ours)	7%	93%	0.5603	0.4117
32 (ours)	7%	93%	0.5709	0.4233
64 (ours)	7%	93%	**0.5781**	**0.4271**

**Table 5 diagnostics-14-01040-t005:** Comparison of acne segmentation results using semi-supervised learning with ACNE04’s pseudo ground truth.

Methods	Ratio		Metrics	
	Labeled	Unlabeled	Dice Score	Jaccard Index
BCP [17]	3%	97%	0.5103	0.3642
Ours	3%	97%	**0.5159**	**0.3702**
BCP [17]	7%	93%	0.5664	0.4141
Ours	7%	93%	**0.5749**	**0.4212**

## Data Availability

Data are contained within the article.

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
