# Peer review of "Semi-Supervised Facial Acne Segmentation Using Bidirectional Copy–Paste"

_diagnostics, 2024, doi:10.3390/diagnostics14101040_

Round 1

Reviewer 1 Report

Comments and Suggestions for Authors

In the study, researchers performed Semi-Supervised Facial Acne Segmentation using Bidirectional Copy-Paste. The work is interesting and novel. However, eliminating various problems found in the study will increase the quality and readability of the study.

1. There is no binding force in the abstract. While acne was mentioned in the first sentence, suddenly what was done in the study was mentioned. What is the purpose of this study? What is the problem? Why was DL applied to this problem? These should be mentioned.

2. [1] [2] [3] [4] references in line 17 should be arranged as [1-4].

3. HSV, SVM, CNN expansions should be given.

4. There are also shortcomings in the introduction section. What are the traditional approaches used to identify acne? What are the problems with these approaches? Why are artificial intelligence approaches used? What kind of solution does it provide to the problems caused by traditional approaches? These questions should be answered and the intro shoudl be updated.

5. The CT expansions on line 49, Faster-RCNN on line 71, and R-FCN on line 72 should be given.

6. Studies in the literature are mentioned under the title "2. Related Works". An introductory sentence should be given before scanning the literature.

7. While the explanation of Figure 2 is given under the title of Method, Figure 2 is displayed under Related Works. The shape must be relocated to the Method section.

8. Figure 3 and Table 4 are not referenced in the article.

9. The previous expansion of BCP is given earlier. There is no need to give it again in lines 110 and 166.

10. The meaning of SGD given in line 155 should be given.

11. Training and loss graphs of the model should be given and interpreted. In this way, it can be checked whether the model is overfit or underfit.

12. There was no discussion. What are the advantages and disadvantages of the study? What are the highlights of this study? What problems might this study have?

13. Conclusion should be improved.

14. The similarity rate was determined as 18%. It should be at most 15%.

15. I haven't come across any sentences written in ChatGPT. However, if researchers have used ChatGPT (there is no problem with this), they should be careful with their sentences and not use ChatGPT's sentences directly.

Comments on the Quality of English Language

Minor english language issues detected.

Author Response

Thank you for taking the time to review our paper and for providing many comments to improve the logic and quality of the manuscript. We have done our best to incorporate your comments and have revised the paper accordingly. Therefore, we have written responses to all your comments as follows, and these have been reflected in the revised manuscript.

General Comments: In the study, researchers performed Semi-Supervised Facial Acne Segmentation using Bidirectional Copy-Paste. The work is interesting and novel. However, eliminating various problems found in the study will increase the quality and readability of the study.
A. We appreciate your thorough review of our paper and will incorporate your comments to improve the manuscript.

  1. Q. There is no binding force in the abstract. While acne was mentioned in the first sentence, suddenly what was done in the study was mentioned. What is the purpose of this study? What is the problem? Why was DL applied to this problem? These should be mentioned.
    A. Thank you for your comments on the necessity of the proposed method. As you suggested, we have revised the paper to improve the flow where it was lacking. Currently, many methods based on deep learning detect acne but often struggle due to the lack of training data. Therefore, we aim to improve this by proposing an acne segmentation method based on semi-supervised learning. Reflecting this, we have added and modified the content from line 2 to line 7 in the revised Abstract as follows.
    (line 2 to 7)
    However, it is important to detect acne early as the condition can worsen if not treated. For this purpose, deep learning-based methods have been proposed to automate detection, but acquiring acne training data is not easy. Therefore, this study proposes a novel deep learning model for facial acne segmentation utilizing a semi-supervised learning method known as bidirectional copy-paste, which synthesizes images by interchanging foreground and background parts between labeled and unlabeled images during the training phase.
  1. Q. [1] [2] [3] [4] references in line 17 should be arranged as [1-4].
    A. I have reflected it in line 19 of the revised manuscript.
    (line 19)
    Facial skin disorders commonly occur among people, which is why various studies are being conducted to detect them [1– 4].
  1. Q. A HSV, SVM, CNN expansions should be given.
    A. In lines 30, 31, and 39 of the revised manuscript, I first wrote the full words and then indicated the abbreviations.
    (line 30) hue-saturation-value (HSV)
    (line 31) support vector machine (SVM)
    (line 39) convolutional neural network (CNN)
  1. There are also shortcomings in the introduction section. What are the traditional approaches used to identify acne? What are the problems with these approaches? Why are artificial intelligence approaches used? What kind of solution does it provide to the problems caused by traditional approaches? These questions should be answered and the intro shoudl be updated.
    A. Thank you for your comments highlighting the necessity of the proposed method. Traditional methods identify acne based on color and texture. However, these methods struggle to account for the diverse shapes and colors of acne, and especially defining acne features as descriptors is difficult for humans to do manually. Therefore, recent acne detection methods have been utilizing deep learning. However, applying deep learning requires a large number of labeled data, which is not easily obtainable. Hence, there is a need for semi-supervised learning methods to improve performance from limited data. The main purpose of this paper is to enhance acne segmentation performance by improving semi-supervised learning methods. Reflecting this, lines 25 to 34 of the revised manuscript introduce traditional methods and summarize their limitations.
    (line 25 to 34)
    Budihi et al. [5] applied the region growing method based on pixel color similarity in facial skin images to select candidate areas for acne. Additionally, a self-organizing map was used to diagnose acne. Alamdari et al. [6] used techniques such as K-means clustering, texture analysis, and color-based segmentation to segment acne areas. Yadav et al. [7] identified candidate regions of acne presence on the face based on the hue-saturation-value (HSV) color space and then distinguished acne using classifiers trained with support vector machine (SVM). However, these methods struggle with setting threshold values for classifying acne’s color or texture. Moreover, defining a descriptor that reflects the diverse characteristics of acne accurately is challenging for humans.
  1. Q. The CT expansions on line 49, Faster-RCNN on line 71, and R-FCN on line 72 should be given.
    A. In lines 30, 31, and 39 of the revised manuscript, I first wrote the full words and then indicated the abbreviations.
    (line 36) faster region-based convolutional neural network (Faster-RCNN)
    (line 37) region-based fully convolutional networks (R-FCN)
    (line 61) computed tomography (CT)
  1. Q. Studies in the literature are mentioned under the title "2. Related Works". An introductory sentence should be given before scanning the literature.
    A. Thank you for your comment suggesting adding descriptions to the sections, which seems to have improved the flow of the paper. Therefore, we have added brief descriptions to the beginning of not only Section 2 but also Sections 3 and 4 before they are divided into subsections.
    (line 94)
    In this section, we briefly review existing acne detection methods based on deep learning and semi-supervised learning approaches to overcome the challenge of insufficient labeling.
    (line 133)
    In this section, we provide a detailed explanation of the proposed method. First, we present the overall structure and explain the basic principles of BCP. Then, we show how the training loss is calculated in the proposed method.
    (line 184)
    In this section, we analyze the performance of the proposed method for acne segmentation and compare it with previous semi-supervised learning methods. First, we describe the experimental setup and then proceed to compare the acne segmentation performance with existing semi-supervised methods.
  1. Q. While the explanation of Figure 2 is given under the title of Method, Figure 2 is displayed under Related Works. The shape must be relocated to the Method section.
    A. Figure 2 was moved to page 4, where it is first mentioned.
  1. Q. Figure 3 and Table 4 are not referenced in the article.
    A. I have reviewed the entire paper and corrected the citations for Figures and Tables that were not mentioned.
  1. Q. The previous expansion of BCP is given earlier. There is no need to give it again in lines 110 and 166.
    A. After line 51, where BCP is first mentioned, I referred to it as BCP without using the full word.
  1. Q. The meaning of SGD given in line 155 should be given.
    A. In line 194 of the revised manuscript, I have written the full word.
    (line 194): stochastic gradient descent (SGD)
  1. Q. Training and loss graphs of the model should be given and interpreted. In this way, it can be checked whether the model is overfit or underfit.
    A. I have added the following at the top of page 9 of the revised manuscript.

  1. Q. There was no discussion. What are the advantages and disadvantages of the study? What are the highlights of this study? What problems might this study have?
    A. In line 265 of the revised manuscript, the Discussion section was added. Throughout this revision, we generated PSEUDO GT on the public ACNE04 database and trained the acne segmentation model in Section 5. However, the ACNE04 used in the experiments mainly consists of Asian skin and has similar ambient lighting. In this case, the performance improvement of the proposed method compared to the previous BCP was reduced. Therefore, while the proposed method may not significantly improve performance in cases of similar skin tones, which can be seen as a drawback, in real-world use, skin tones are very diverse due to various ethnicities. Thus, the proposed method is considered superior to the existing BCP in practical applications. This related content has been written from line 272 to 283 of the revised manuscript.
    (line 272 to 283)
    The proposed method showed improved performance on our data compared to the original BCP method. However, as observed in the ablation study in section 5, there was a decrease in acne segmentation performance improvement in ACNE04. To analyze this, we compared synthetic images composed from each dataset as depicted in Figure 7. Our data includes a variety of skin colors and diverse lighting conditions, while the ACNE04 dataset used in experiments is predominantly composed of East Asian skin tones. Therefore, even when creating synthetic images using ACNE 04, as shown in Figure 7, the images composited in the foreground exhibit less disparity and lower color gradation compared to those composed with our data. Nevertheless, actual human skin tones vary widely in color depending on race and environmental conditions. Therefore, it is expected that our method will demonstrate superior performance in actual acne segmentation compared to the orginial BCP approach.
  1. Q. Conclusion should be improved.
    A. Thank you for your comments to improve the conclusion section of this paper. We have reread the conclusion and made the following modifications to enhance the quality of the manuscript.
    (line 285 to 296)
    In this paper, a semi-supervised learning method for training acne segmentation models is proposed. The original BCP method, which calculated the loss solely on synthetic images, was insufficient for detecting acne across diverse skin tones. To address this, the training process was enhanced by including original images to improve overall acne segmentation performance. The proposed method was compared with previous semi-supervised learning methods on acne patch images and demonstrated superior performance based on evaluation metrics such as the Dice score and Jaccard index. However, the performance improvement was less significant in ACNE04, where skin color and lighting are similar. Since actual human skin color and lighting vary, performance improvements like those in the main results of this paper are expected in real applications. Future research aims to apply the proposed method across the medical imaging field to enhance performance in various areas.
  1. Q. The similarity rate was determined as 18%. It should be at most 15%.
    A. Typically, professors and students use Turnitin for plagiarism checks, but as a company researcher, it's difficult for me to use that program. Instead, I've used QuillBot's plagiarism checker to assess the plagiarism rate. As shown in the figure below, the current rate is 5.9%, which is lower than the 15% you mentioned. However, the plagiarism program used by the reviewer may be different. Nonetheless, due to the additional content added through revisions, I predict that using the plagiarism program you mentioned will result in a rate below 15%. For reference, my paper is currently published on Preprints.org to prevent idea theft, so you can exclude this paper from the evaluation. (https://www.preprints.org/manuscript/202404.0591/v1)
  1. Q. I haven't come across any sentences written in ChatGPT. However, if researchers have used ChatGPT (there is no problem with this), they should be careful with their sentences and not use ChatGPT's sentences directly.
    A. As you suggested in your comment, we will take care when using ChatGPT.

Reviewer 2 Report

Comments and Suggestions for Authors

The paper proposes a bidirectional copy-paste (BCP) based semi-supervised learning method for Facial Acne Segmentation. The paper is well-structured and easy to follow. There are some suggestions,

1.     The technical novelty of the work comes from the combination of several existing techniques. The only contribution here is the use of synthetic images which are created using cut and paste.

2.     Please add ablation study on the dataset with and without synthetic loss to validate the use of synthetic images.

3.     The authors should conduct experiments on a publicly available dataset to show the effectiveness of the proposed approach. https://paperswithcode.com/dataset/acne04

4.     Inclusion of pseudo code will be beneficial to the readers.

5.     Discussion on limitations and failure cases will add value to the paper.

6.     Please remove results from introduction.

7.     From Figure 4., it is unclear what are the different columns, please add subheadings.

8.     The authors should explain the choice of using U-Net-BN in the proposed approach. Is there any ablation study conducted?

9.     Authors mentioned “Although BCP performed well with CT images, we discovered shortcomings in segmenting acne across various shapes and skin tones.” However, this is not detailed in the experiments.

10.  Look at the recent published paper on the same topic: https://www.mdpi.com/2673-7426/4/2/59

Comments on the Quality of English Language

Correct the grammatical mistakes and typos. 

Author Response

Thank you for taking the time to review our paper and for providing many comments to improve the logic and quality of the manuscript. We have done our best to incorporate your comments and have revised the paper accordingly. Therefore, we have written responses to all your comments as follows, and these have been reflected in the revised manuscript.

Reviewer 2

General Comments: The paper proposes a bidirectional copy-paste (BCP) based semi-supervised learning method for Facial Acne Segmentation. The paper is well-structured and easy to follow. There are some suggestions,

  1. A. Thank you for reviewing our paper. As you suggested in your comments, we have reflected them as much as possible to improve the quality of the paper.
  2. Q. The technical novelty of the work comes from the combination of several existing techniques. The only contribution here is the use of synthetic images which are created using cut and paste.
    A. Thank you for your comments aimed at enhancing the main contributions of our paper. As you mentioned, the main idea is the structure that combines synthetic loss with labeled loss in the BCP method. We conducted an ablation study to fuse synthetic loss and labeled loss. We also validated the effectiveness of the proposed method on the public database ACNE04. Additionally, we mentioned the performance differences based on skin color in our database and ACNE04 in the discussion. Furthermore, we summarized the performance based on the number of parameters in U-Net. To better organize our contributions, we have reflected the following in the revised manuscript.
    (line 78 to 86)
    The main contributions of the proposed method are as follows:
    Fusion of labeled loss and synthetic loss: We propose a method that simultaneously calculates and fuses the labeled loss for training labeled images and the synthetic loss for training unlabeled images.
    • Comparison of acne segmentation performance with semi-supervised learning methods: We compared the acne segmentation performance with previous semi-supervised learning methods based on our acne database and ACNE04. Additionally, through ablation studies, we compared the acne segmentation performance as the parameters of U-Net were increased.
  1. Q. Please add ablation study on the dataset with and without synthetic loss to validate the use of synthetic images.
    A. We apologize for any confusion in the presentation of our paper. Synthetic loss is used for training unlabeled data, while pre-trained weights are trained by computing labeled loss with labeled images. Therefore, the absence of synthetic loss refers to the stage of training pre-trained weights. It seems that the training method was not clearly conveyed, possibly due to the absence of a comprehensive pseudo code, as indicated in comment 4. Thus, we have added Algorithm 1 to include the pseudo code (I will further explain this in comment 4). Additionally, we changed 'Method' to 'Loss Type' in Table 1 and removed the 'Pre-trained' section in Table 2. Also, Sub-section 3.2 has been added to specify that before semi-supervised learning, the pre-trained weights are trained using supervised learning.
  1. Q. The authors should conduct experiments on a publicly available dataset to show the effectiveness of the proposed approach. https://paperswithcode.com/dataset/acne04
    A. The ACNE04 you mentioned does not have segmentation ground truth. However, it does have bounding boxes for acne for object detection purposes. Although not precise, we drew circles within the bounding boxes to create pseudo ground truth. After removing low-quality images with text and selecting only those taken under similar conditions, we were able to choose 1,108 images of East Asians. Using these, we conducted experiments in the same manner as in Section 4 for the ablation study. However, we observed a slight reduction in the performance improvement in acne segmentation in our database. Our database consists of a variety of skin colors from different ethnicities and diverse lighting conditions, whereas ACNE04 mostly consists of images taken under identical lighting and of the same ethnicity. This result is related to comment 9, indicating that the more similar the color distribution of the training data, the smaller the performance improvement of the proposed method, and conversely, the greater the diversity in color distribution, the larger the performance improvement. In the revised manuscript, we have included the method of generating labels and the related experimental results as follows.

  1. Q. Inclusion of pseudo code will be beneficial to the readers.
    A. Thank you for your comment to enhance the understanding of the paper. We have included the proposed method as a pseudo code algorithm in the revised manuscript. This algorithm will likely aid in comprehending the paper. Below is the algorithm included in the revised manuscript.

  1. Q. Discussion on limitations and failure cases will add value to the paper.
    A. Based on your comments, we have revised the paper and created a discussion section to address the limitations of the proposed method. As mentioned earlier, one drawback is that the performance improvement of the proposed method decreases as the skin color and surrounding lighting become more similar. However, in real-world use, where various skin tones and lighting conditions exist, we believe the proposed method will be more beneficial. We have written the following in the revised manuscript.
    (line 272 to 283)
    The proposed method showed improved performance on our data compared to the original BCP method. However, as observed in the ablation study in section 5, there was a decrease in acne segmentation performance improvement in ACNE04. To analyze this, we compared synthetic images composed from each dataset as depicted in Figure 7. Our data includes a variety of skin colors and diverse lighting conditions, while the ACNE04 dataset used in experiments is predominantly composed of East Asian skin tones. Therefore, even when creating synthetic images using ACNE 04, as shown in Figure 7, the images composited in the foreground exhibit less disparity and lower color gradation compared to those composed with our data. Nevertheless, actual human skin tones vary widely in color depending on race and environmental conditions. Therefore, it is expected that our method will demonstrate superior performance in actual acne segmentation compared to the original BCP approach.

  1. Q. Please remove results from introduction.
    A. As you mentioned, I removed the numerical results from the introduction.

  1. Q. From Figure 4., it is unclear what are the different columns, please add subheadings.
    A. Thank you for pointing out our mistake. In Figure 4, we added lines to separate each image and represented what each image means with subheadings.

  1. Q. The authors should explain the choice of using U-Net-BN in the proposed approach. Is there any ablation study conducted?
    A. I apologize for the confusion. The U-Net-BN we mentioned is identical to the one used in the referenced BCP. However, we denoted it as U-Net-BN because the original U-Net published in 2015 did not include batch normalization, and our different notation seemed to cause more confusion. Therefore, we have reverted the notation back to U-Net and indicated this with a citation.
  2. Q. Authors mentioned “Although BCP performed well with CT images, we discovered shortcomings in segmenting acne across various shapes and skin tones.” However, this is not detailed in the experiments.
    A. As mentioned in comment 3, we have confirmed that when the skin tone becomes more uniform through ACNE04, the extent of performance improvement of BCP and the proposed method decreases, whereas the performance improvement increases when the skin tone becomes more diverse. Therefore, we would like to add the following to the revised manuscript and discuss it in the discussion section.
    (line 66 to 68)
    Additionally, in the ablation study and discussion, we conducted acne segmentation experiments using ACNE04 [22], which has similar skin tones, and analyzed issues related to this.

  1. Q. Look at the recent published paper on the same topic: https://www.mdpi.com/2673-7426/4/2/59
    A. Thank you for providing a helpful paper on performance improvement. In fact, we are also very interested in enhancing acne detection performance using generative AI. We have briefly mentioned another paper related to this topic in the main text.
    (line 54 to 55)
    Additionally, to overcome the lack of training data, it was proposed to use generative models like StyleGAN2 [18 ][19 ] to create images for use in acne training [20 ][21 ].

    Added references
    [20] Sankar, A., Chaturvedi, K., Nayan, A., Hesamian, M., Braytee, A. & Prasad, M. Utilizing Generative Adversarial Networks for 351 Acne Dataset Generation in Dermatology. BioMedInformatics. 4, 1059-1070 (2024,4) 352
    [21] Kim, S., Lee, J., Lee, C. & Lee, J. Improving Facial Acne Segmentation through Semi-Supervised Learning with Synthetic Images. 353 Journal Of Korea Multimedia Society. 27, 241-249 (2024,2)

Round 2

Reviewer 1 Report

Comments and Suggestions for Authors

The researchers made the relevant revisions and updated the article. The article is acceptable in its current form.